# High Fructose Corn Syrup Accelerates Kidney Disease and Mortality in Obese Mice with Metabolic Syndrome

**DOI:** 10.3390/biom13050780

**Published:** 2023-04-30

**Authors:** Ana Andres-Hernando, David J. Orlicky, Christina Cicerchi, Masanari Kuwabara, Gabriela E. Garcia, Takahiko Nakagawa, Laura Gabriela Sanchez-Lozada, Richard J. Johnson, Miguel A. Lanaspa

**Affiliations:** 1Division of Endocrinology, Metabolism and Diabetes, University of Colorado Anschutz Medical Campus, Aurora, CO 80045, USA; 2Division of Nephrology, Rocky Mountain VA Medical Center, Aurora, CO 80045, USA; 3Department of Pathology, University of Colorado Anschutz Medical Campus, Aurora, CO 80045, USA; 4Division of Cardiovascular Disease, Toranomon Hospital, Tokyo 105-8470, Japan; 5Division of Renal Diseases and Hypertension, University of Colorado Anschutz Medical Campus, Aurora, CO 80045, USA; 6Department of Regenerative Medicine Development, Shiga University of Medical Science, Seta Tsukinowa-cho, Otsu 520-2192, Japan; 7Department of Cardio-Renal Physiopathology, INC Ignacio Chávez, Mexico City 14080, Mexico

**Keywords:** metabolic syndrome, obesity, chronic kidney disease, fructose, high fructose corn syrup

## Abstract

The presence of obesity and metabolic syndrome is strongly linked with chronic kidney disease (CKD), but the mechanisms responsible for the association are poorly understood. Here, we tested the hypothesis that mice with obesity and metabolic syndrome might have increased susceptibility to CKD from liquid high fructose corn syrup (HFCS) by favoring the absorption and utilization of fructose. We evaluated the pound mouse model of metabolic syndrome to determine if it showed baseline differences in fructose transport and metabolism and whether it was more susceptible to chronic kidney disease when administered HFCS. Pound mice have increased expression of fructose transporter (Glut5) and fructokinase (the limiting enzyme driving fructose metabolism) associated with enhanced fructose absorption. Pound mice receiving HFCS rapidly develop CKD with increased mortality rates associated with intrarenal mitochondria loss and oxidative stress. In pound mice lacking fructokinase, the effect of HFCS to cause CKD and early mortality was aborted, associated with reductions in oxidative stress and fewer mitochondria loss. Obesity and metabolic syndrome show increased susceptibility to fructose-containing sugars and increased risk for CKD and mortality. Lowering added sugar intake may be beneficial in reducing the risk for CKD in subjects with metabolic syndrome.

## 1. Introduction

Metabolic syndrome is common in the United States and affects approximately one-third of the adult population [1]. By definition, metabolic syndrome is a cluster of conditions that include at least three of the following features: fatty liver, insulin resistance, high blood pressure, obesity, and dyslipidemia. Metabolic syndrome is a major risk factor for cardiovascular disease, stroke, and type 2 diabetes. It is also a major risk factor for chronic kidney disease (CKD), even before hypertension or diabetes develops [2]. For subjects with none of the criteria for metabolic syndrome, only 0.3% have CKD (defined as stage 3 or higher) compared to 9.2 percent who carry all five criteria (i.e., 30-fold difference) [2]. We have previously reported that in a genetic murine model of metabolic syndrome associated with leptin resistance and hyperphagia, obese mice spontaneously develop CKD as they age and that if CKD is induced by a toxin (adenine), it will progress faster towards end-stage kidney disease (ESKD) [3]. Mice with metabolic syndrome develop not only CKD but also sarcopenia and have a reduced lifespan [3]. Thus, a better understanding of both what causes metabolic syndrome and why metabolic syndrome is associated with CKD is of great clinical importance.

Our group and others have identified sugar and high fructose corn syrup (HFCS) as major drivers of metabolic syndrome [4,5]. Both of these added sugars contain fructose, which we have identified as a nutrient that uniquely increases the risk of inducing metabolic syndrome through its ability to suppress mitochondrial function and lower intracellular adenosine triphosphate (ATP) levels [6,7]. The reduction in intracellular ATP acts as an alarm signal to orchestrate a series of biological effects to aid survival, including inducing hunger, foraging, leptin resistance, lipogenesis, fat accumulation, elevations in blood pressure, and insulin resistance [8]. One of the biological actions of fructose is also to increase glomerular hydrostatic pressure, likely as a mechanism to facilitate urinary excretion [9]. Fructose also stimulates vasopressin production [10]. In turn, chronic fructose-induced alterations in glomerular pressure and vasopressin are associated with both glomerular and tubular injury in laboratory rats [9,11,12]. There are clinical studies linking sugar and HFCS intake with both CKD [13,14,15] as well as increased mortality in patients with CKD [16].

On the other hand, humans being overweight, obese, or metabolically unhealthy independently increase the risk of developing CKD. Furthermore, the composite of overweight/obesity and metabolic abnormalities significantly increases such risk [17].

These studies raised the question of the effect of fructose intake on the development and progression of CKD in animals with and without metabolic syndrome. We hypothesized that animals with metabolic syndrome might be at greater risk for developing CKD in response to fructose, especially if administered an HFCS-containing soft drink. To study this, we utilized the pound mouse, a mouse with a genetic loss of the leptin receptors that markedly predisposes the mouse to develop massive obesity and metabolic syndrome.

## 2. Materials and Methods

### 2.1. Study Approval

All animal experiments were conducted with adherence to the National Institutes of Health Guide for the Care and Use of Laboratory Animals. The animal protocol was approved by the Institutional Animal Care and Use Committee of the University of Colorado (Aurora, CO, USA).

### 2.2. Animal Experiments 

Pound mice with a C57BL/6NCrl genetic background were initially isolated in a Charles River Laboratories barrier facility; this strain is made obese by the deletion of the leptin receptor. Pound and wild-type (WT) littermates with a C57BL/6 background were obtained from Charles River (Wilmington, MA, USA). KHK-A/C KO (B6;129-Khk^tm2Dtb^) mice were originally developed by David Bonthorn at Leeds University (UK) and were bred and maintained at the Univ. Colorado for over seven generations to ensure the mice were on the B6 genetic background. All experimental mice (female and male) were maintained in temperature- and humidity-controlled specific pathogen-free conditions on a 14-h dark/10-h light cycle and allowed ad libitum access to normal laboratory chow (Harlan Teklad, #2920X). Caloric intake was calculated by measuring daily food (Teklad 2920X, 3.1 Kcal/g) and sugar water intake (4 kcal/g with varying % of HFCS depending on the strain, wild type (10%) or KHK-A/C KO (18%)). Then, the total or fructose-derived cumulative caloric intake was calculated as the cumulated sum of daily chow and/or sugar water intake multiplied by the number of days in the study. Importantly, as body weights were markedly different, the calculated dosing in oral fructose and FITC dextran assays was normalized by lean mass. To this end, fat and fat-free (lean) mass was determined by Echo-MRI. All experiments were conducted with adherence to the National Institutes of Health Guide for the Care and Use of Laboratory Animals. The animal protocol was approved by the Animal Care and Use Committee of the Veterans Affairs Medical Center. In all experiments, 8-week-old male mice were used.

### 2.3. Oral Fructose Tolerance Test 

For portal vein and circulating fructose studies, animals were gavaged orally with fructose at a dose of 1.25 g/kg corrected by lean mass. Portal veins and circulating blood were collected on mice at the indicated times under isoflurane anesthesia, and serum was obtained after centrifugation at 13,000 rpm for 2 min at room temperature. Fructose levels were determined biochemically following the manufacturer’s instructions (bioassay systems, EFRU-100).

### 2.4. Intestinal Permeability Assay 

For the studies investigating intestinal permeability in obese and lean mice, lean (fat-free mass) was first calculated with a EchoMRI Body Composition Analyzer (EchoMRI, Houston, TX, USA) 24 h prior to the study. On the study day, mice were fasted for 5 h before fluorescein isothiocynate (FITC)-Dextran 4 kDA administration (600 mg/kg dissolved in PBS at a concentration of 125 mg/mL) via oral gavage and blood collected 45′ post challenge. A volume of 150 μL of plasma was analyzed with a fluorescence spectrophotometer (520 nm) using a Promega GloMax + plate reader along with a standard dilution of FITC-dextran.

### 2.5. Tissue KHK Activity 

Kidney, gut mucosa, and liver samples were first homogenized in 20 mM Tris-HCl, pH 7.5, 150 mM KCl, 1 mM EDTA, and 1 mM DTT using a polytron homogenizer, and centrifuged for 10 min at 13,000 rpm at 4 °C. The protein content of the supernatant fraction was quantified with the protein BCA assay, and Khk activity was measured on 50 μg lysate protein after the addition of a buffer to 5 mM fructose in 50 mM imidazole, 1 M potassium acetate, pH 5.2, and 1 mM ATP. ATP was measured both before and after a 2-h incubation at 37 °C using the ATP determination kit (K354-100, Biovision) as per the manufacturer’s instructions. Khk activity was calculated as the ratio between ATP levels at 2 h versus baseline for each sample at zero time.

### 2.6. Biochemical Analysis 

Blood was collected in Microtainer tubes (BD) from cardiac puncture of mice under isoflurane, and serum was obtained after centrifugation at 13,000 rpm for 2 min at room temperature. Serum parameters were performed biochemically following the manufacturer’s instructions [blood urea nitrogen (BUN): DIUR-100, Bioassay Systems, Hayward, CA, USA; uric acid: DIUA-250, Bioassay Systems; Creatinine: C753291, Pointe Scientific; fructose: EFRU-100, Bioassay Systems]. Urinary neutrophil gelatinase-associated lipoprotein (NGAL; MLCN20, R&D Systems, Minneapolis, MN, USA), and albumin (Albuwell M; Ethos Biosciences) levels were normalized to units of creatinine. Determination of parameters in tissue was performed in freeze-clamped tissues and measured biochemically following the manufacturer’s protocol [uric acid: DIUA-250, Bioassay Systems; Hydroxyproline: DHYP-100, Bioassay Systems; thiobarbituric acid-reactive substances (TBARS): DTBA-100, Bioassay Systems; and uric acid: DIUA-250, Bioassay Systems].

### 2.7. Histopathology 

Formalin-fixed paraffin-embedded intestinal and kidney sections were stained with periodic acid-Schiff (PAS). Histological examination was performed through the entire cross section of the kidney from each mouse. Images were captured on an Olympus BX51 microscope equipped with a 4-megapixel Macrofire digital camera (Optronics, Goleta, CA, USA) using the PictureFrame Application 2.3 (Optronics). Composite images were assembled with the use of Adobe Photoshop. All images in each composite were handled identically. Kidney score was performed as described in Table 1 analyzing pathology in four major segments, renal corpuscle, cortex, medulla, and other pathological findings. Kidneys were stained with Picro Sirius Red to assess fibrosis as previously described. Images were captured on an Olympus BX51 microscope equipped with a 4-megapixel Macrofire digital camera (Optronics, Goleta, CA, USA) using the PictureFrame Application 2.3 (Optronics) and using polarized light. Composite images were assembled with the use of Adobe Photoshop. All images in each composite were handled identically.

### 2.8. Western Blot Analysis 

Protein lysates were prepared from mouse tissue using MAPK lysis buffer as previously described [18]. Protein content was determined by the BCA protein assay (Pierce, Rockford, IL, USA). Total protein (50 μg) was separated by SDS-PAGE [10% (wt/vol)] and transferred to PVDF membranes (Bio-Rad, Hercules, CA, USA). Membranes were first blocked for 1 h at 25 °C in 4% (wt/vol) instant milk dissolved in 0.1% Tris-buffered saline with Tween 20 TBS (TTBS) and incubated with the following primary rabbit- or mouse-raised antibodies (1:1000 dilution in TTBS): KHK (HPA007040, Sigma, St. Louis, MO, USA), Glut5 (07-1406, Millipore, Burlington, MA, USA), and actin (no. 3700, Cell Signaling) and visualized using an anti-rabbit (no. 7074) or anti-mouse IgG (no. 7076) horseradish peroxidase (HRP)-conjugated secondary antibody (1:2000, Cell Signaling) using the SuperSignal West Pico PLUS (34577, ThermoFisher Scientific). Chemiluminescence was recorded with an Image Station 440CF, and results were analyzed with the 1D Image software (Kodak Digital Science, Rochester, NY, USA). Data for proteins of interest are normalized to β-actin expression.

### 2.9. DNA Isolation and Quantitative Real-Time PCR 

For mitochondrial DNA number, total (nuclear and mitochondrial) DNA was extracted from renal tissues (A1120, Promega, Madison, WI, USA), following the recommended protocol. Mitochondria number was considered as the ratio between mitochondrial and nuclear DNA and by determining DNA levels of a mitochondrial protein codified by mtDNA (COXII) against the number of copies of another mitochondrial gene codified by nuclear DNA, nDNA (UCP2). The TaqMan 7900 HT system was used to perform real-time PCR amplification of the mtDNA regions using the following primers and probes: Forward Primer for Cox II: 5′-aattgctatcccctctcacg-3′, Reverse Primer for Cox II: 5′-gtagcttcattggtgc-3′, Forward Primer for UCP2: 5′- agcctacaagaccattgcacgaga-3′, Reverse Primer for UCP2: 5′-ataggtcaccagctcagcacagtt-3′. All primers were obtained from Integrated DNA Technologies Inc. (Coralville, IA, USA). The real-time PCR assays were performed in triplicate for each sample. qPCR was performed in TaqMan 7900 HT Fast Real-time PCR System with primer concentrations of 500 nM. Cycling variables were 5 min at 95 °C and then 54 cycles of 10 s at 95 °C and 1 min at 60 °C.

### 2.10. Statistical Analysis 

All numeric data are presented as means ± SE. Independent replicates for each data point (*n*) are shown in the figures. Data graphics and statistical analysis were performed using Prism 5 (GraphPad). Data without indications were analyzed by one-way ANOVA with a Tukey post hoc test. *p* values of <0.05 were regarded as statistically significant.

## 3. Results

### 3.1. Metabolic Syndrome Is Associated with Leaky Gut and Enhanced Fructose Absorption

To determine the potential effects of obesity on fructose uptake and metabolism, we employed pound mice, a mouse strain with hyperphagia, and increased caloric intake as a consequence of deficient leptin signaling [3,7]. At 8 months of age, a clean difference in fat mass (Figure 1A) and body weight (56.3 ± 5.7 g vs. 34.2 ± 3.7 g, *p* < 0.01) was observed between pound obese and lean littermates. Of interest, obese mice showed a 20–40% increase in intestinal villus length in the duodenum and proximal jejunum compared with lean counterparts (Figure 1B,C) and greater intestinal permeability to a paired dose (normalized to lean mass) of fluorescein-labelled dextran (FITC-Dextran) (Figure 1D), indicating that the pound mouse has a higher gut absorptive area associated with increased gut leakiness.

Fructose metabolism in the intestinal epithelium has been reported to efficiently increase villus length and expand the surface area of the gut [19] while increasing intestinal permeability [20]. We, therefore, examined if fructose absorption and metabolism might be increased in obese mice compared to lean mice. We measured fructose levels and clearance in the portal vein and systemic circulation following a single fructose oral gavage (1.25 g/kg, corrected for lean body mass) in lean and obese mice never exposed to fructose before (naïve). As shown in Figure 1E,F, obesity was associated with greater intestinal fructose absorption at 10′ and 15′ post-fructose administration (2.3 ± 0.4x increase in portal vein fructose, *p* < 0.01) as well as with greater circulating levels of fructose associated with reduced clearance. Consistent with greater extra-splanchnic fructose availability, the expression of the main fructose transporter (Glut5) in the gut was significantly up-regulated in obese mice despite not being exposed to fructose before (Figure 1G,H). Similarly, the expression of ketohexokinase (KHK, also known as fructokinase), the first enzyme involved in fructose metabolism [21,22], was also significantly up-regulated in the liver and kidneys of obese mice (Figure 1G,H) compared to lean counterparts suggesting increased fructose processing in these organs.

### 3.2. Chronic Intake of a HFCS-like Beverage Accelerates Development of CKD in Obese Mice 

To determine whether higher fructose intestinal absorption rates and metabolism would be indicative of a much greater susceptibility of obese mice to the deleterious effects of sugar, we then analyzed the effect to long-term exposure to clinically relevant doses of a fructose–glucose beverage in lean and pound mice. To this end, a glucose–fructose (45/55 ratio) solution similar to high fructose corn syrup (HFCS) was provided in the drinking water of obese mice at a concentration (10%) similar to those found in soft drinks. As shown in Figure 2A,B, pre-administration of HFCS for 2 weeks further elevated portal vein and systemic levels of fructose following a fructose oral gavage compared to weight-matched naïve obese but not lean mice. We previously have demonstrated that obesity accelerates kidney disease in mice by causing metabolic dysregulation in the proximal tubule, which was associated with a significant reduction in the survival rate of obese pound mice with lifespans of less than 12 months [3].

The administration of HFCS to obese mice resulted in significantly earlier mortality compared to obese controls (9.2 ± 2.3 months in HFCS-exposed obese vs. 14.6 ± 2.9 in obese control mice, Figure 2C). In contrast, mortality was similar in lean mice independent of whether they received HFCS. The reduced lifespan observed in obese mice receiving HFCS was associated with significantly worse kidney damage that was observed as early as 6 months. Specifically, blood urea nitrogen (BUN), plasma creatinine, and albuminuria were relatively similar in lean mice with or without HFCS and control obese control mice; kidney dysfunction was significantly greater in HFCS-exposed obese mice (Figure 2D). HFCS-fed obese mice at 6 months of age also had worse kidney injury scores characterized by proximal tubular dilatation, reduced brush border expansion, and the presence of features of metabolic dysregulation and tubular injury including glycosylated nuclei and increased urinary excretion of NGAL (Figure 2E–G). Kidney function continued to worsen at 12 months in surviving mice, especially in obese mice receiving HFCS in association with higher levels of KHK activity in the kidneys (Figure 2H,I).

### 3.3. Blockade of Fructose Metabolism Protects against HFCS-Induced Kidney Dysfunction and Injury in Obese Mice 

To determine the importance of fructose metabolism in sugar-induced kidney dysfunction in obese mice, we generated pound mice deficient for both main isoforms of KHK (KHK-A and KHK-C, namely KHK-A/C KO) and analyzed their response to HFCS. Baseline analysis between pound obese and obese-KHK-A/C KO mice revealed similar age-dependent weight gain (body weight at 8 months was 52.5 ± 2.7 g in obese-KHK-A/C KO vs. 55.3 ± 4.6 g in wild-type obese littermates) and adiposity (41.3 ± 4.1% fat mass in obese-KHK-A/C KO vs. 46.2 ± 2.8% in wild type obese littermates). However, consistent with previous reports in KHK-A/C KO mice [23,24], HFCS intake was significantly reduced in obese-KHK-A/C KO compared with wild-type obese counterparts (5.7 ± 1.2 mL/day in obese-KHK-A/C KO vs. 11.3 ± 2.6 mL/day in wild type obese littermates, *p* < 0.01). Based on this significant difference in HFCS intake, we administered 10% HFCS to obese mice and gave 18% HFCS solution to obese-KHK-A/C KO mice to provide equivalent HFCS, and we did this for 6 to 18 months. This way, we matched both sugar and caloric intake as well as fructose-derived calories (Figure 3A). No differences in weight were observed at 8 months of age between HFCS-exposed obese-KHK-A/C KO and wild-type obese counterparts (Figure 3B). However, despite similar adiposity between groups, the survival rate observed in pound mice on HFCS was significantly improved when KHK expression was deleted (10.5 ± 2.6 months in HFCS-exposed obese vs. 17.4 ± 1.2 in HFCS-exposed obese-KHK-A/C KO mice, Figure 3C), indicating that the specific blockade of fructose metabolism in mice improves the lifespan of obese mice in response to HFCS in the drinking water. Importantly, at 12 months of age, the increased survival rate in HFCS-exposed obese-KHK-A/C KO mice was associated with a marked improvement in kidney function as noted by lower plasma BUN (82.7 ± 8.1 mg/dL in HFCS-exposed obese vs. 54.3 ± 8.4 mg/dL in HFCS-exposed obese-KHK-A/C KO mice), creatinine (0.78 ± 0.05 mg/dL in HFCS-exposed obese vs. 0.47 ± 0.08 mg/dL in HFCS-exposed obese-KHK-A/C KO mice), and albuminuria (92 ± 19.8 μg/mg in HFCS-exposed obese vs. 66.3 ± 9.5 μg/mg in HFCS-exposed obese-KHK-A/C KO mice) (Figure 3D–F), less histologic injury (Figure 3G,H and Table 1), and lower urinary NGAL levels (43.9 ± 18.1 μg/mL in HFCS-exposed obese vs. 16.3 ± 6.7 μg/mL in HFCS-exposed obese-KHK-A/C KO mice, Figure 3I). Injury scoring in this study is described in detail in Table 1. Specifically, chronic exposure of obese mice to HFCS demonstrated particular pathological features in the renal corpuscle (hypercellularity, protein casts in glomeruli, thickened basal membrane), cortex (tubular casts, glycogenated nuclei, peritubular and peri arcuate inflammation, and pigmented macrophages), medulla (cast and inflammation) as well as inflammation in peri-interlobar artery. Furthermore, kidney fibrosis, as denoted by both cortical levels of hydroxyproline and tubulointerstitial collagen staining with picrosirius red, was also significantly higher in obese mice on HFCS compared to obese-KHK-A/C KO mice receiving the same amount of HFCS (Figure 3J,K).

### 3.4. Fructose Metabolism Controls Fructose Transport in the Kidney

Circulating fructose is filtered by kidney glomeruli and reabsorbed from the lumen in the proximal tubule by the fructose transporter Glut5 (SLC2a5), where it is then metabolized via KHK in the proximal tubule [12,25]. Much of the kidney injury in response to fructose results from the proximal tubule metabolism associated with oxidative stress and inflammation [25]. We found that renal Glut5 expression was significantly reduced in obese-KHK-A/C KO mice on water or exposed to HFCS compared to obese littermates (44.3 ± 5.3% reduction in HFCS-exposed obese-KHK-A/C KO mice compared to HFCS-exposed obese mice, *p* < 0.01, Figure 4A,B) indicating that renal fructose metabolism is necessary for Glut5 expression and fructose uptake in the kidney. Consistent with this finding, urinary fructose excretion was elevated in obese-KHK-A/C KO mice (17.8 ± 7.4 nmol/mg in HFCS-exposed obese vs. 330.8 ± 88.8 nmol/mg in HFCS-exposed obese KHK-A/C KO mice, *p* < 0.01) (Figure 4C), and this was associated with less metabolic dysfunction in the kidneys compared to obese controls receiving HFCS (Figure 4D–I). Of interest, the loss of fructose in the urine in obese KHK-A/C KO mice was similar to that of lean KHK-A/C KO counterparts receiving consuming similar amounts of HFCS (286.8 ± 76.8 nmol/mg). While kidneys of HFCS-exposed obese mice had significantly reduced energy charge (total nucleotide levels, Figure 4D) and higher oxidative stress as noted by higher levels of thiobarbituric reactive substances (TBARS) and uric acid (Figure 4E,F), it was significantly less in obese-KHK-A/C KO receiving HFCS. Similarly, higher mitochondrial oxidative stress (noted by elevated mitochondrial superoxide production detected with dihydroethidium) (Figure 4G,H) and lower proximal tubule mitochondrial number (Figure 4I) were present in obese mice receiving HFCS, while the obese-KHK-A/C KO receiving HFCS were protected.

## 4. Discussion

Here, we tested the hypothesis that consumption of fructose-containing added sugars might accelerate the development of CKD and overall mortality in obese mice with metabolic syndrome. The model we used was the pound mouse, which has a mutation in the leptin receptor, leading to hyperphagia and all of the features of the metabolic syndrome [26]. We had several novel observations. First, we found that the pound mouse with obesity and metabolic syndrome showed enhanced fructose transport and metabolism, along with a leaky gut, even before exposure to HFCS. Second, when the obese mouse was administered HFCS, it showed enhanced absorption and developed worse obesity associated with CKD and increased mortality rates. Third, in KHK-KO pound mice, the development of kidney disease and early mortality was prevented despite the mice still gaining excess weight. The preserved kidney function was associated with the downregulation of the fructose transporter in the proximal tubule (Glut5), decreased intrarenal oxidative stress and uric acid levels, and less mitochondrial oxidative stress and injury. Overall, these studies document the critical role of fructose in driving kidney damage in animals with obesity.

Our first finding was that the obese pound mouse had evidence for enhanced fructose metabolism, as noted by higher levels of the intestinal transporter, Glut5, as well as higher liver fructokinase (KHK) activity, along with higher portal and systemic blood levels of fructose following administration of fructose (based on lean mass) compared to its lean littermate. One of the major regulators for Glut5 and KHK is fructose (such as present in sucrose or HFCS) [27], but this upregulation was observed in mice despite them having not previously been exposed to dietary fructose, our study used fructose-naïve animals. The upregulation of this pathway may be due to the hyperuricemia that is observed in the pound mouse, as increased uric acid can stimulate both aldose reductase in the polyol pathway [28] as well as KHK [29]. In turn, the increase in endogenous fructose would likely have a role in stimulating the growth of the intestinal villi [19] as well as disrupting the tight junctions leading to a leaky gut [30]. Thus, the pound mouse is not only hyperphagic from disrupted leptin signaling but also shows enhanced sensitivity to fructose, a larger intestinal absorptive area, and gut leakiness that may enhance weight gain and inflammation.

The second finding was that when pound mice were placed on HFCS, they showed an increased risk for not only obesity (Figure 3B) but also with greater risk for CKD and for mortality than lean mice placed de novo on HFCS or compared to obese pound mice that did not receive HFCS. Excessive fructose administration is known to cause kidney injury and has been previously reported to accelerate kidney injury in animals with CKD [11,12]. While one mechanism appears to be mediated by an increase in glomerular hydrostatic pressure with a reduction in cortical blood flow [9,31], a major mechanism involves direct tubular injury related to the metabolism of fructose by KHK in the proximal tubule [25]. This is associated with mitochondrial dysfunction, local oxidative stress, and inflammation [12,25,32]. The striking finding in this study was that the injury was much more severe, leading to marked azotemia and reduced survival. The reason appears to be from the marked sensitivity to fructose, as noted by higher expression of the fructose transporter (Glut5) and higher KHK activity associated with higher fructose absorption. Pound mice also became fatter on HFCS despite the underlying inactivation of leptin signaling. While gaining fat and developing CKD, lean mice did not develop the same degree of kidney disease and maintained typical survival rates. Thus, having metabolic syndrome from the start made the pound mice especially sensitive to the effects of added sugars such as HFCS. The observation that added sugars may shorten lifespan has also been observed in humans [33], including in patients with CKD [16].

The final major finding was that the increased mortality rates and CKD induced by HFCS were largely prevented in the KHK-KO obese pound mice. The mechanism was shown to not only simply block the metabolism of fructose by KHK in the kidney but also block the uptake of fructose in the proximal tubule by downregulating the Glut5 receptor. This was an interesting observation as it may explain why fructosuria is so prominent in subjects lacking KHK (essential fructosuria). The reduced uptake of fructose was associated with the preservation of mitochondria, reduced mitochondrial and intracellular oxidative stress, and normalization of uric acid levels in the kidney. These studies emphasize the key role of fructose in driving kidney damage and implicate fructose as one of the mechanisms by which metabolic syndrome may cause CKD. Interestingly, knocking out KHK in the pound mouse tended to correct the weight gain associated with the addition of HFCS but did not do so completely. This finding further emphasizes the overriding importance of leptin resistance in driving weight gain [34]. In addition, it is important to note that renal dysfunction and injury in obese KHK-A/C KO tend to be mildly ameliorated compared to obese wild-type controls even in the absence of HFCS. This would indicate the presence of active endogenous production and metabolism of fructose (as opposed to that provided by the HFCS) in the kidney of obese mice. In this regard, we and others have shown that the endogenous production of fructose and its metabolism via fructokinase in states of diabetes or ischemia are important deleterious steps in the pathogenesis of acute [35] and chronic [36] kidney disease.

In conclusion, our studies suggest that the effects of fructose are amplified in mice with obesity and metabolic syndrome (Figure 5). This amplification is due to the upregulation of the fructose transporter, increased relative absorption rates, and higher metabolism. The sum of these metabolic adaptations translates into developing CKD more rapidly in association with a lower lifespan. The take-home message is that the intake of added sugars is especially hazardous in obese subjects with metabolic syndrome, as it significantly increases the risk of developing CKD.

Our study has several limitations. Unlike a more generally used model of diet-induced obesity, here, we employ a genetic model of obesity in which leptin signaling is curtailed causing hyperphagia and greater caloric intake. We preferred to use this model as opposed to for example a high-fat (western diet) diet as these diets are generally rich in sugar (up to 34%) and therefore, fructose. Thus, the interpretation of how specifically important liquid sugar is in accelerating kidney disease in obese mice would be more difficult if sugar is also provided with the chow. In addition, it allowed us to utilize both male and female mice in this study and to have a more cohesive group of animals with minimal standard deviation. However, the blockade of leptin signaling itself may also exert deleterious effects on renal function and injury independently of body weight or obesity. For example, hyperleptinemia is observed in subjects with chronic kidney disease [37] and leptin levels increase with its progression [38,39]. Another limitation of using the pound mouse is that it may not be the best genetic background (B6) to study chronic kidney disease. The B6 background demonstrates only modest susceptibility to kidney disease on its own, and it normally requires experimental intervention such as nephrectomy, DOCA-Salt, angiotensin II, or in our case liquid sugar, which has a greater deleterious effect in obese mice. Other genetic backgrounds such as the 129, FVB and DBA/2 strains are more susceptible to kidney disease, and therefore, in these strains, liquid sugar could accelerate kidney disease not only in obese but also in lean mice on these strains. Another limitation is that our study does not elucidate the site or organ where fructose metabolism is important to accelerate kidney disease in obese mice and the specific isoform of KHK involved in the process. In this regard, there are two major isoforms of KHK, KHK-A and KHK-C. Of these, KHK-A is a ubiquitous protein with a low affinity for fructose, while KHK-C is expressed primarily in the gut, liver, and kidney and has a high affinity for fructose. Previously, we demonstrated the importance of KHK-C as the main isoform driving fructose-derived metabolic dysregulation and disease [24,36,40,41]. This is in part due to its high affinity for fructose, which causes acute depletion of ATP and the activation of the purine degradation pathway and uric acid formation. High urinary fructose in obese KHK-A/C KO mice would suggest that it is KHK-C activation that drives the disease in obese mice, but a study employing KHK isoform-specific deficient mice is needed. Similarly, our study will not elucidate the site where fructose is metabolized to induce kidney disease in obese mice. The presence of no major differences in weight gain between obese wild type and obese KHK-A/C KO mice on HFCS would suggest that it is the local metabolism of fructose in the kidney that accelerates kidney dysfunction in obese mice, but future studies employing tissue-specific and isoform-specific obese mice are thus warranted.

## Figures and Tables

**Figure 1 biomolecules-13-00780-f001:**
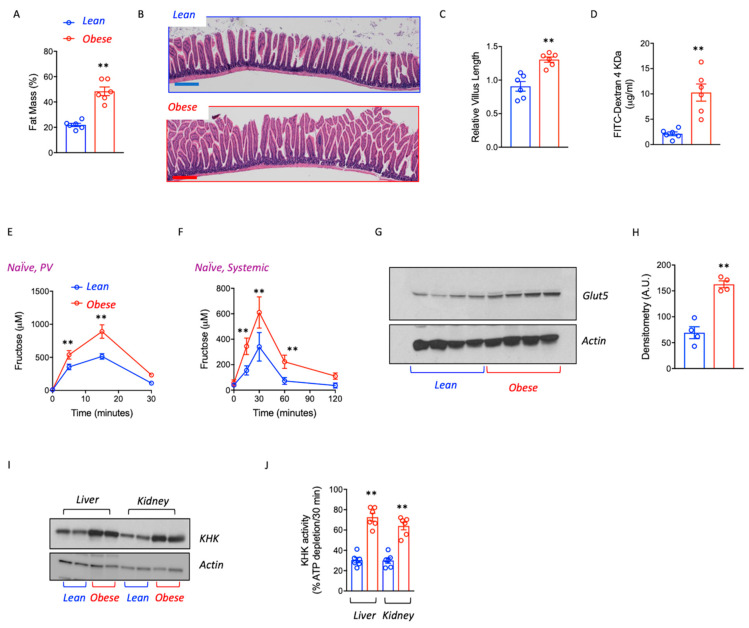
Obesity favors fructose utilization and metabolism. (**A**) Fat mass percentage in naïve (never exposed to fructose) and similar age lean and obese mice. (**B**) Representative image of intestinal histology. Scale bar 200 μm. (**C**) Relative jejunal villus length in lean and obese mice. (**D**) Intestinal permeability measured as circulating levels of FITC-Dextran 4 kDa in lean and obese mice. (**E**) Portal vein fructose levels in lean and obese mice challenged with a 1.25 g/kg oral fructose gavage. (**F**) Systemic circulating fructose levels in lean and obese mice challenged with a 1.25 g/kg oral fructose gavage. (**G**) Glut5 and GAPDH expression in the gut of lean and obese mice. (**H**) Densitometry analysis of Western blot expression in (**G**). (**I**) KHK-A/C (KHK), and actin control expression in the liver and kidney cortex and Glut5 and actin in kidney and gut of lean and obese mice. (**J**) KHK-A/C (KHK) activity in liver and kidney cortex of lean and obese mice. *n* = 6 mice per group. The bar graphs show mean ± SEM. ** *p* < 0.01 versus lean mice by two-tail *t*-test analysis.

**Figure 2 biomolecules-13-00780-f002:**
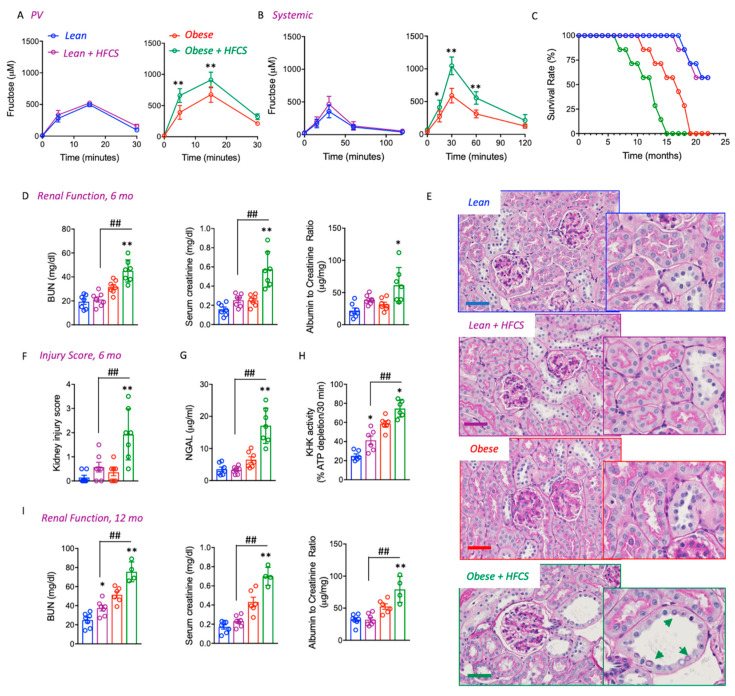
HFCS Accelerates CKD and Mortality in Obese Mice. (**A**) Portal vein fructose levels in naïve (never exposed before, red) or chronically fed high fructose corn syrup (HFCS, green) lean and obese mice challenged with a 1.25 g/kg of lean mass oral fructose gavage. (**B**) Systemic circulating fructose levels in naïve (never exposed before, red) or chronically fed high fructose corn syrup (HFCS, green) lean and obese mice challenged with a 1.25 g/kg of lean mass oral fructose gavage. (**C**) Survival rate in lean (blue), lean fed HFCS (purple), obese control (red) and obese fed HFCS (green) over a 22-month period (**D**) Renal function determined as plasma creatinine (left), blood urea nitrogen (BUN, center), and albuminuria (right) in lean (blue), lean fed HFCS (purple), obese control (red), and obese fed HFCS (green) at 6 months of age. (**E**) Representative PAS kidney images of lean (blue), lean fed HFCS (purple), obese control (red), and obese fed HFCS (green) at 6 months of age. Green arrows indicate the glycosylated nuclei in, dilated distal tubules. Scale bar 200 μm (**F**) Injury score in lean (blue), lean fed HFCS (purple), obese control (red) and obese fed HFCS (green) at 6 months of age. (**G**) Urinary NGAL in lean (blue), lean fed HFCS (purple), obese control (red), and obese fed HFCS (green) at 6 months of age. (**H**) KHK-A/C (KHK) activity in kidney cortex of lean (blue), lean fed HFCS (purple), obese control (red), and obese fed HFCS (green) at 12 months of age. (**I**) Renal function determined as plasma creatinine (left), blood urea nitrogen (BUN, center), and albuminuria (right) in lean (blue), lean fed HFCS (purple), obese control (red), and obese fed HFCS (green) at 12 months of age. *n* = 4–7 mice per group. The bar graphs show mean ± SEM. * *p* < 0.05 and ** *p* < 0.01 versus respective lean or obese control mice, ^##^ *p* < 0.01 between obese and lean mice on HFCS by one way ANOVA and post-hoc Tukey’s comparison test.

**Figure 3 biomolecules-13-00780-f003:**
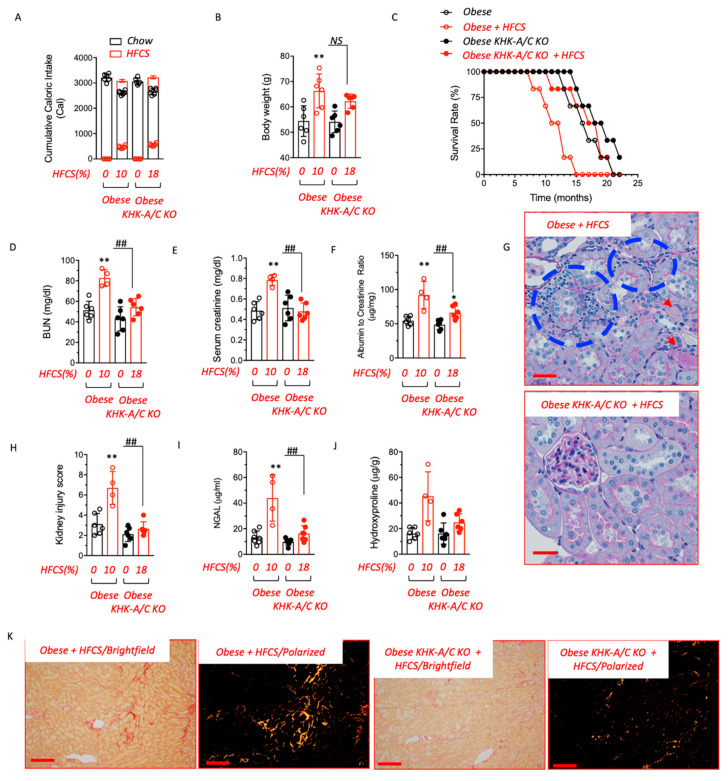
Blockade of fructose metabolism protects against HFCS-induced CKD in obese mice. (**A**) The 10-month cumulative caloric intake from chow (black) and high fructose corn syrup (HFCS, red) in pair-matched obese wild type (obese) and KHK-A/C knockouts (KO). (**B**) Body weight at 10 months of age of the same mice as in (**A**). (**C**) The survival rate of obese control and KHK-A/C knockouts on water (black) or fed HFCS (red) over a 22-month period. (**D**–**F**) Renal function determined as plasma creatinine (left), blood urea nitrogen (BUN, center), and albuminuria (right) in obese control and KHK-A/C knockouts on water (black) or fed HFCS (red) at 12 months of age. (**G**) Representative PAS kidney images of obese control and KHK-A/C knockouts fed HFCS at 12 months of age. Blue areas denote inflammatory foci. Red arrows show tubular cast formation. Scale bar 200 μm. (**H**) Injury score in obese control and KHK-A/C knockouts on water (black) or fed HFCS (red) at 12 months of age. (**I**) Urinary NGAL in obese control and KHK-A/C knockouts on water (black) or fed HFCS (red) at 12 months of age. (**J**) Renal hydroxyproline levels in obese control and KHK-A/C knockouts on water (black) or fed HFCS (red) at 12 months of age. (**K**) Representative picrosirius red (PSR) images on brightfield and polarized light of obese control and KHK-A/C knockouts fed HFCS at 12 months of age. Scale bar 40 μm *n* = 4–7 mice per group. The bar graphs show mean ± SEM. * *p* < 0.05, ** *p* < 0.01 versus respective water control, ^##^ *p* < 0.01 and *NS* not-significative by one-way ANOVA and post-hoc Tukey’s comparison test.

**Figure 4 biomolecules-13-00780-f004:**
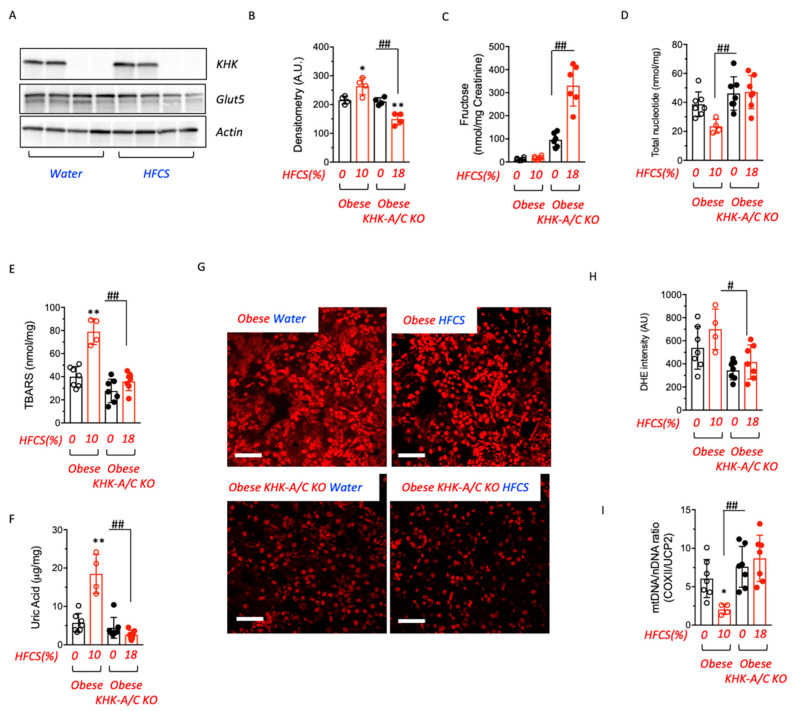
Renal fructose metabolism is necessary for fructose reabsorption and metabolic dysregulation. (**A**,**B**) KHK-A/C (KHK), Glut5 and actin control expression and densitometry in kidney cortex of obese control and KHK-A/C knockouts on water or fed HFCS at 12 months of age. (**C**) 24-h urinary fructose levels in obese control and KHK-A/C knockouts on water (black) or fed HFCS (red) at 12 months of age. (**D**) Energy charge (total nucleotide ATP+ADP+AMP levels) in obese control and KHK-A/C knockouts on water (black) or fed HFCS (red) at 12 months of age. (**E**) Renal thiobarbituric reactive substances (TBARS) in obese control and KHK-A/C knockouts on water (black) or fed HFCS (red) at 12 months of age. (**F**) Renal uric acid in obese control and KHK-A/C knockouts on water (black) or fed HFCS (red) at 12 months of age. (**G**,**H**) Representative dihydroethidium images and quantification of obese control and KHK-A/C knockouts fed water or HFCS at 12 months of age. Scale bar 100 μm. (**I**) Renal mitochondrial (mtDNA) to nuclear DNA (nDNA) ratio in obese control and KHK-A/C knockouts on water (black) or fed HFCS (red) at 12 months of age, COXII cytochrome C Oxidase II, UCP2 Uncoupling protein 2. *n* = 4–7 mice per group. The bar graphs show mean ± SEM. * *p* < 0.05 and ** *p* < 0.01 versus respective water control, ^#^ *p* < 0.05 and ^##^ *p* < 0.01 significative by one-way ANOVA and post-hoc Tukey’s comparison test.

**Figure 5 biomolecules-13-00780-f005:**
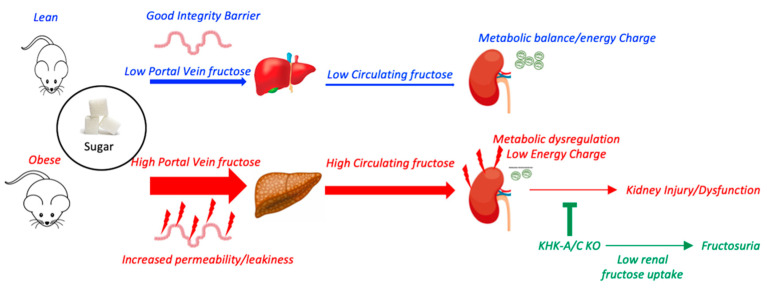
The proposed mechanism whereby sugar exacerbates renal disease in obese mice. Obesity is associated with intestinal dysfunction and increased permeability/leakiness. In these conditions, for the same exposure to sugar, portal veins and systemic levels of fructose are markedly elevated in obese mice in association with higher fructokinase (KHK) expression in the liver and kidney. Furthermore, renal energy charge and mitochondrial function in obese mice are reduced compared to lean counterparts, and therefore, higher fructose delivery and metabolism in the kidney lead to greater metabolic dysfunction, oxidative stress, and inflammation. This effect induced by dietary liquid sugar is markedly ameliorated when renal fructose metabolism is deleted as it impairs glut5-dependent fructose uptake and reabsorption causing fructosuria.

**Table 1 biomolecules-13-00780-t001:** Renal Injury Scoring in lean and obese mice on water or HFCS.

			**Wild Type**		**KHK-A/C KO**		**Pound**		**Pound KHK-A/C KO**	
			**Water**	**HFCS**	**Water**	**HFCS**	**Water**	**HFCS**	**Water**	**HFCS**
**Feature**		**Points**	***n* = 7**	***n* = 7**	***n* = 7**	***n* = 7**	***n* = 6**	***n* = 4**	***n* = 6**	***n* = 6**
(1) Renal Corpuscle										
Mesangial expansion	None	0	x	x	x	x	x	x	x	x
(0–2)	Yes, central	1–2								
(0–2)	Yes, crescent	1–2								
Hypercellularity	None	0	x	x	x	x			x	x
(0–2)	Present	1–2					x	x		
Inflammation	None	0	x	x	x	x	x	x	x	x
(0–2)	Moderate	1								
	Severe	2								
Protein Casts in glomeruli (Vessels)	None	0	x	x	x	x	x		x	x
(0–2)	Present	2						x		
Protein Casts in glomerular urinary space	None	0	x	x	x	x	x	x	x	x
(0–2)	Present	1–2								
if yes, give approx % of RC with this	Check > 25RC	na								
thickened basement membrane	None	0	x	x	x	x	x		x	x
(0–2)	Moderate	1						x		
	Severe	2								
renal corpuscle loss (0–2)	None (<10%)	0	x	x	x	x	x	x	x	x
	Yes (>10%)	2								
afferent arteriole hyalinized/thickened	No	0	x	x	x	x	x	x	x	x
	Yes	1								
metaplasia	Ct Only	Ct								
proxim c metaplas of renal corpus pariet c	<20%	0	x	x	x	x	x	x	x	x
(>180 degree or stratified)	>20%	1								
Bowman’s Capsule thickened	None	0	x	x	x	x	x	x	x	x
(0–2)	Present	1–2								
Renal corpuscle subtotal			0	0	0	0	0.7	2.4	0	0
(2) Cortex										
Acute Tubular Necrosis (0–2)	None	0	x	x	x	x	x	x	x	x
	Moderate	1								
	Severe	2								
Any cortical casts, cell, protein	None	0	x		x	x	x		x	x
(0–2)	Moderate	1		x				x		
	Severe	2								
PCT Pathology (0–2)	None	0	x	x	x	x		x	x	
	Simplification	1					x			x
	Necrosis	2								
PCT vacuolization present	Y/N									
Extent (a=little b=lots)	A,B,C,N									
Tubular Dilatation	None	0	x		x	x				
	Present	1		x			x	x	x	x
DCT Pathology (0–2)	None	0	x	x	x	x	x	x	x	x
	Prot,disrupt	1								
	Necrosis	2								
DCT Glycogen granules (0–1)	None	0	x		x	x	x		x	x
	Present	1		x				x		
Peritub/Interstit Inflamm	None	0							x	
Near level of the AA (0–2)	Moderate	1					x			x
	Severe	2						x		
Increase peritub interstit -tis fibrosis	None	0	x	x	x	x	x		x	x
Near level of the AA (0–2)	Moderate	1						x		
	Severe	2								
Brown Pigmented macs	None	0	x	x	x	x	x		x	x
	Present	1						x		
Hemosiderosis on collect duct	None	0	x	x	x	x	x	x	x	x
	Present	1								
Peri-arcuate artery inflamm	None	0	x	x	x	x	x		x	x
Around 10–50% of AA	Moderate	1						x		
Around >50% of AA	Severe	2								
Cortex Subtotal			0	1.1	0	0	1.8	3.1	0.8	1.6
(3) Medulla										
OM Casts (0–2)	None	0	x	x	x	x				
	Present	1–2					x	x	x	x
IM Casts (0–2)	None	0	x	x	x	x				
	Present	1–2					x	x		x
Inflammation (0–2)	None	0	x	x	x	x			x	x
	Present	1–2					x	x		
Medulla subtotal			0	0	0	0	1.1	1.6	0.7	1.2
(4) Other										
Peri-interlobar artery inflammation	None	0	x	x	x	x	x		x	x
(0–2)	Moderate	1						x		
	Severe	2								
Pelvic Inflammation	None	0	x	x	x	x	x	x	x	x
(0–2)	Moderate	1								
	Severe	2								
Other subtotal			0	0	0	0	0	0.5	0	0
Summary										
**Scoring by Category**			**Wild type**		**KHK-A/C KO**		**Pound**		**Pound KHK-A/C KO**	
			**Water**	**HFCS**	**Water**	**HFCS**	**Water**	**HFCS**	**Water**	**HFCS**
			***n* = 7**	***n* = 7**	***n* = 7**	***n* = 7**	***n* = 7**	***n* = 6**	***n* = 7**	***n* = 7**
(1) Renal Corpuscle			0	0	0	0	0.7	2.4	0	0
(2) Renal Cortex			0	0.5	0	0	1.8	3.1	0.8	1.6
(3) Medulla			0	0	0	0	1.1	1.6	0.7	1.2
(4) Other			0	0	0	0	0	0.5	0	0
Total			0	0.5	0	0	3.6	7.6	1.5	2.8

## Data Availability

Further information and requests for resources and reagents should be directed and will be fulfilled by the Lead Contact, Miguel A. Lanaspa (miguel.lanaspagarcia@cuanschutz.edu). Mouse lines generated in this study are available for any researcher upon reasonable request. This study did not generate unique datasets or code.

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
