# Peer review of "High Fructose Corn Syrup Accelerates Kidney Disease and Mortality in Obese Mice with Metabolic Syndrome"

_biomolecules, 2023, doi:10.3390/biom13050780_

Round 1

Reviewer 1 Report

Given the increasing incidence of obesity and consumption of sugar-sweetened drinks, the study is highly desirable and important. However, I have a couple of suggestions/ questions: 

1.     In Introduction, briefly explain criteria for metabolic syndrome. 

2.     Please provide a new blot for Glut 5 and corresponding housekeeping gene expression in Gut in figure 1. Also, quantative graph is missing. 

3.     In Figure 2, statistics is not clear. What difference ## is showing?

4.     In Figure 2E, provide higher magnification image for all groups. 

5.     Why was injury score not calculated for 12 months in figure 2. 

6.     Data not shown for lines 162-169. 

7.     How was cumulative calorie intake was calculated or monitored/ mice?

8.     In Figure 3 I and J, why only 4 mice in obese + HFCS group, especially when you have higher number of mice for other measurements. 

9.     Table heading is missing and not explained in detail in text. 

10.  Figure 4A, label obese vs KO. Figure 4B, specific GLUT 5 in y-axis. 

11.  What was the urine level of fructose in lean mice? Was it comparable to obese KHK-A/C KO + HFCS group. 

12.  There is a decrease in TBARS, DHE intensity in KHK-A/C KO group even with no fructose.  Does this changes happen independent of fructose stimulation. 

None

Reviewer 2 Report

The manuscript of Ana Andres-Hernando et al. entitled: “High Fructose Corn Syrup Accelerates Kidney Disease and Mortality in Obese Mice with Metabolic Syndrome” attempted to evaluate the pound mouse model of metabolic syndrome to determine if it showed baseline differences in fructose transport and metabolism and whether it was more susceptible to chronic kidney disease when administered HFCS.

My comments are listed below:

1)Methods section entitled “Western blot analysis” provides only some general background For mitochondrial DNA number assessment and lacks basic information, such as used primers for real-time PCR by quantifying or efficiency for Ct of gene of interest or reference gene. Similarly, the quality of the (Fig. 4I) is insufficient.

2) Authors conclude in Discussion section “Pound mice also became fatter on HFCS despite the underlying inactivation of leptin signaling”. Please clarify what other results could be expected and what are the limitations of the study.

3) Tissue KHK activity was measured. Can Authors specify the role of KHK-A/C and discuss if this study is consistent with data presented in manuscript “Opposing effects of fructokinase C and A isoforms on fructose-induced metabolic syndrome in mice" by Takuji Ishimotoa et all.,

4) Please specify what are the limitations of the study. Such as importance to sequence the genomes of B6 mouse strain.
